# A Human Behaviour Perspective on Horizontal Collaboration to Reduce the Climate Impact of Logistics

**Frans Cruijssen** [1,*] , **Ilja van Beest** [2] and **Goos Kant** [1]

1   Department of Econometrics and Operations Research, Tilburg School of Economics and Management, Tilburg University, Warandelaan 2, 5037 AB Tilburg, The Netherlands; g.kant@tilburguniversity.edu
2   Department of Social Psychology, Tilburg School of Social and Behavioural Sciences, Tilburg University, Warandelaan 2, 5037 AB Tilburg, The Netherlands; i.vanbeest@tilburguniversity.edu
*   Correspondence: frans.cruijssen@tilburguniversity.edu

**Abstract:** The transport sector needs to drastically reduce its carbon footprint to comply with the Paris Agreement. In today's sharing economy, an emerging strategy to contribute to this goal is horizontal collaboration. However, most studies on horizontal collaboration or resource pooling are approached from a theoretical Operations Research perspective, and case studies are usually stylised. At the same time, the uptake of horizontal collaboration in practice is limited. An important explanation for this is that compared to traditional vertical collaboration, coalition formation is much more complex under horizontal collaboration, as some players will be included in the collaborative coalition, while others will be excluded. We conjecture that this renders human behaviour much more important than in more traditional vertical supply chain relations. Therefore, in this paper, we propose a research agenda for an interdisciplinary approach that integrates human behavioural aspects in studies on horizontal supply chain collaboration. We review some vital concepts from social psychology and discuss the importance to the success or failure of horizontal collaboration initiatives to reduce the environment footprint of the logistics sector. We conclude that social psychological insights on mixed-motive interactions are pivotal to understand wicked problems such as Sustainable Development Goal 13 on Climate Action, and that interdisciplinary approaches should therefore receive more attention in academic literature.

**Keywords:** horizontal collaboration; social psychology; coalition formation; interdisciplinarity; climate action; sustainable development; resource pooling

## 1. Introduction

Efficient transport is fundamental to our economy and society. The International Transport Forum [1] expects that freight transport activity will grow 2.6-fold by 2050 compared to 2015, to support expected economic growth and respond to trends in consumer preferences. Under the current ways of operating transport networks, however, this rapid growth will result in a strong growth of carbon emissions. Ceteris paribus, in 2050, the absolute carbon emissions from freight transport will be 22% higher than in 2015. This obviously conflicts with the recent stringent environmental legislation and policies, such as the European Union's effective ban on the sale of new petrol and diesel cars and trucks from 2035 onwards. Given that the transport sector is responsible for one quarter of global greenhouse gas emissions, it is imperative that the industry transforms in order to contribute to the achievement of Sustainable Development Goal 13 of Climate Action.

### 1.1. Horizontal Collaboration in Transport

Fortunately, reducing the environmental footprint of freight transport is not an impossible challenge. In fact, absolute freight emissions can be reduced by 72% [1] under effective policies to:

- Boost freight consolidation;
- Enhance collaboration and resource pooling in supply chains;
- Advance standardisation;
- Promote low-carbon technologies across the sector.

In this paper, we will focus on the second item from this list: enhancing collaboration and resource pooling in supply chains.

Horizontal collaboration has received increasing attention in academic literature [2] and is defined as "collaboration between two or more firms that are active at the same level of the supply chain and perform a comparable logistics function" [3]. Traditionally, this topic has been studied through the lens of a single discipline, mostly within the field of Operations Research (OR), which does not do right by the multi-disciplinary nature of the wicked problem (see Section 2.2) of reducing carbon emissions of the freight transport industry.

A literature search on the terms "horizontal collaboration/cooperation" and "supply chain" in the 2000–2022 period showed that the number of publications on horizontal collaboration as a means to improve efficiency and reduce carbon emissions is steadily on the rise and has seen a sharp growth since 2020 (see Figure 1).

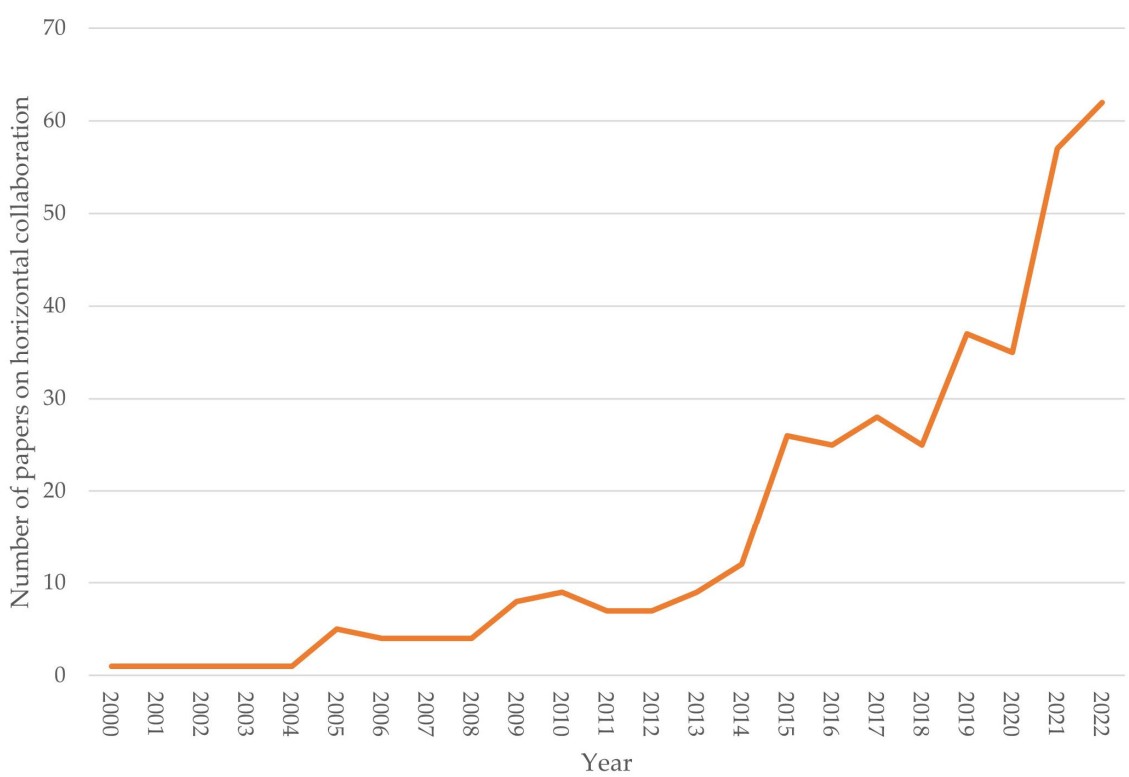

**Figure 1.** Peer-reviewed papers on horizontal collaboration in the 2000–2022 period.

The growing attention for horizontal supply chain collaboration in academia is further illustrated by review papers that have appeared with increasing frequency over the last two decades. The first academic review was provided by Cruijssen et al. [3] and was followed by Verdonck et al. [4], Gansterer and Hartl [2], Pan et al. [5] and most recently by Aloui et al. [6]. These last authors show that the top five journals based on the number of papers published on horizontal supply chain collaboration have a strong emphasis on methodological analysis: 1. *European Journal of Operational Research*; 2. *Transportation Research Part E: Logistics and Transportation*; 3. *Journal of Cleaner Production*; 4. *Computers & Operations Research*; and 5. *Expert Systems with Applications*. This illustrates that the topic is mostly approached from an operations research-based analytical perspective. This means that horizontal collaboration is usually understood through the lens of rationality,

or prescriptive normative models predicting how people should maximise their benefits, leaving whether people indeed behave in accordance with such models unanswered.

### 1.2. The Limited Success of Horizontal Collaboration in Practice

Gansterer and Hartl [2] investigated the advantages of horizontal collaborations in vehicle routing and concluded that most authors find (potential) transport cost savings of 20–30%. Although largely theoretical, these considerable savings make it understandable that horizontal collaboration is considered an important potential contributor to reducing the carbon emissions of freight transport [1]. Unfortunately, in practice, only very few horizontal collaboration initiatives result in long-lasting reductions in ton kilometres and carbon emissions that are in line with the calculated savings beforehand. From the perspective of a modeler, this is an anomaly.

The disappointing results of horizontal collaboration achieved in real life relative to the modelled savings beforehand can be due to various reasons. Quantitative analysts will typically return to their models in search of incorrect data or modelling mistakes. However, a different explanation is that the classical quantitative approach to estimate the expected benefits of horizontal collaboration disregards crucial aspects that do not relate to quantifiable potential savings per se, but to human behaviour.

### 1.3. Goal of the Research

The goal of this paper is to establish an interdisciplinary research agenda that encourages the quantitative research community studying horizontal collaboration to explicitly model human behaviour in order to enhance the logistics and supply chain organisations' sustainable capabilities. To do so, the key concepts from social psychology that impact the adoption and lasting success of horizontal collaboration in logistics are explained. The definition of social psychology by Allport [7] as "the scientific attempt to understand and explain how the thought, feeling, and behaviour of individuals are influenced by the actual, imagined, or implied presence of others" makes it clear that human behaviour is an important determinant of the success of a horizontal collaboration project.

Integrating these concepts into the design of horizontal collaboration will increase the probability of sustained success of horizontal collaboration initiatives. We explicitly choose not to provide clear-cut solutions on how exactly to model these concepts into OR models to make them more realistic in practice. This would require zooming in on only one or a few selected concepts from social psychology. Instead, in this paper, we provide a broad overview of key concepts that are generally disregarded in the literature on horizontal collaboration, but arguably have a strong impact on the probability of the success of a horizontal collaboration initiative. We hope to spark academic interest to enrich horizontal collaboration models in this manner.

The remainder of this paper is organised as follows. Section 2 discusses the connection between OR and social psychology. Then, in Section 3, behavioural concepts from social psychology and their impact on horizontal collaboration are explored. Finally, in Section 4, conclusions are formulated, and directions for further research are proposed.

## 2. Materials and Methods

In this section, we provide the preliminaries to explain how our study builds upon earlier published literature. All literature that has been used can be found in the list of citations at the end of the paper and there are no specific restrictions to report on the availability of this material.

Andersen et al. [8] state that economists typically underestimate the importance of theories from psychology. Decision making models may benefit from the inclusion of elements such as (different) personality traits, personal preferences, cognitions and the interactions of decision makers with their environment (cf. [9,10]).

We investigated how often psychology is mentioned in papers in the *European Journal of Operational Research* (*EJOR*), the journal in which most papers on horizontal collaboration

were published [6]. Figure 2 shows the absolute number and percentage of research papers in *EJOR* since 2000 that mention psychology. On average, 2% of the papers discussed psychology, which is relatively low compared to other fields such as law (14%), government (14%) and marketing (12%). Furthermore, most EJOR papers that do contain the word "psychology" only casually mention it as a context variable, without carefully integrating psychological insights into the optimisation model itself. In short, it appears that this nexus between OR and psychology has indeed received little attention. Although a purely quantitative approach is perfectly fit to analyse well-defined and structured optimisation problems, such as the Vehicle Routing Problem, we posit that insights from other disciplines are crucial when optimisation problems arise in a fuzzy real-life context, such as horizontal collaboration.

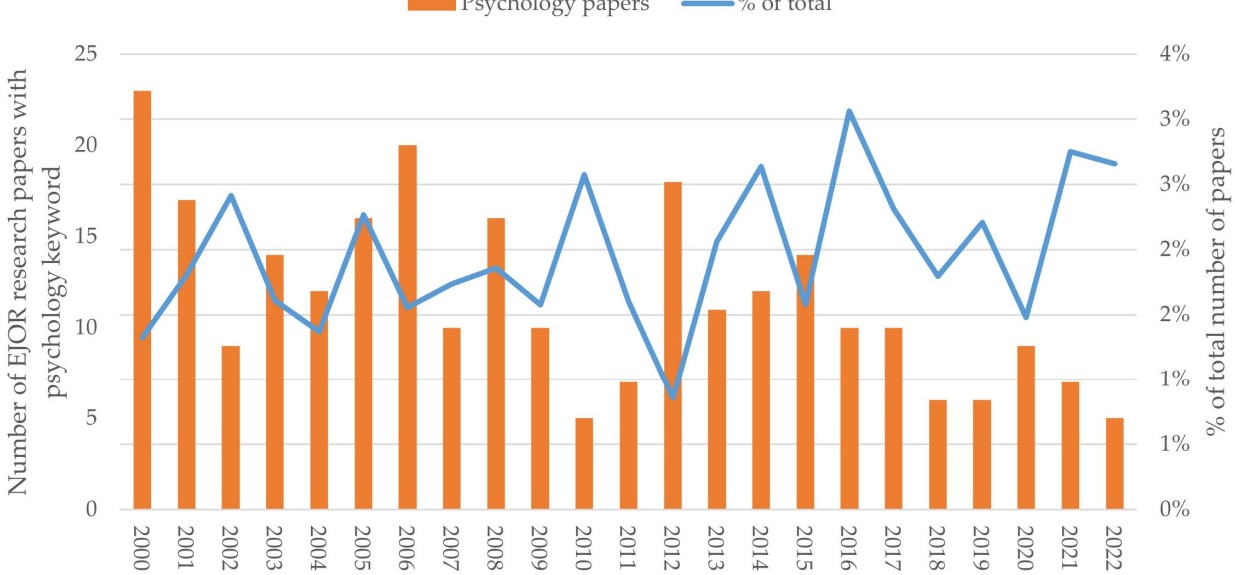

**Figure 2.** Number of research papers in EJOR per year that mention psychology.

### 2.1. Interdisciplinary Operations Research

Interdisciplinary research involves researchers from more than one discipline that come together ad hoc to solve a complex problem [11], and is regarded as one of the strengths of early OR [12]. For example, the 1964 conference on OR and the Social Sciences aimed at broadening the scope of OR interventions through the inclusion of social scientists in so-called mixed OR project teams. However, interdisciplinary research has not spread widely within the OR community since. This brings the risk of tunnel vision and a gap between OR theory and the challenges that supply chain leaders face in reality. This problem of tunnel vision can be alleviated if expertise from other fields is included in the modelling, i.e., interdisciplinary research, as it increases the stimulation of novel approaches and decreases the risk of neglecting important variables.

### 2.2. Wicked Problems

The reduction in carbon emissions of the transport industry in general, and the uptake of horizontal collaboration in particular, is an apparent interdisciplinary challenge and can be regarded as a so-called wicked problem. Churchman [13] introduced the concept of wicked problems as "ill-formulated social problems, with confusing information, multiple clients and decision-makers with conflicting values, where the ramifications for the whole system are thoroughly confusing". He concluded that OR cannot solve these completely, but only a portion of them.

The criteria for judging the validity of a "solution" to a wicked problem are strongly stakeholder-dependent. For example, with horizontal collaboration, one of the first and key



decisions to make is the coalition of companies to collaborate with. Decision-makers have a virtually infinite set of potential partners. Social psychology explains partner selection in coalition formation by explaining the psychological effect that inclusion in, or exclusion from, a group has on individuals. Indeed, stressing the social aspects of coalition formation, social psychologists define coalition formation not only as (a) the process in which two or more agents join forces to achieve a coveted goal that they would not be able to achieve on their own, but also as (b) a process in which agents who achieve the formation of a coalition determine not only the outcomes of those included, but also those who are excluded from the coalition.

Instead of using a purely financial frame, a social frame might be a useful addition to increase adoption and success. In contrast to Operations Research, the field of marketing has realised the usefulness of taking a more social approach. In addition to formal decision models, researchers embrace concepts from social psychology for framing products, services or concepts and nudging consumers in such a way that they do not want to miss out.

Unfortunately, academic literature on horizontal collaboration still largely takes a purely OR perspective [2] and, as a result, accepts strong abstractions of the actual challenge that decision-makers at companies face. There is an implicit assumption of rational decision-making within the strictly demarcated context of the model. We argue that much can be gained when the wicked problem of achieving horizontal collaboration in transport is approached in an interdisciplinary manner and the mostly stylised OR models of horizontal collaboration are developed into richer problems.

### 2.3. Soft Operations Research: The Beer Distribution Game

Including elements from other disciplines in OR models is sometimes referred to as 'soft OR' [14]. Soft OR allows for a range of distinct views and embraces multiple, sometimes conflicting, objectives, without trying to collapse them into a single-objective function. It encourages the active participation of various stakeholders and aims for exploration and learning rather than optimisation only, which makes it possible to focus on the usefulness of the solution and its real-world feasibility [15].

A widely studied example where hard modellers and soft OR experts meet is the classical beer distribution game, an educational game that is often used to illustrate coordination problems in a supply chain process. It is a simulation game that can be played online or as a tabletop game where four participants manage a traditional vertical supply chain without information sharing or other coordination. The game was designed to illustrate the bullwhip (or: demand amplification) effect, but it also nicely lends itself to behavioural experiments. Yang et al. [16] provide a systematic literature review on the bullwhip effect from a behavioural operations perspective, reviewing 53 academic studies on this topic. They demonstrate the importance of understanding human factors in the bullwhip effect and suggest that future studies should consider social and cultural influences on decision making in studying the bullwhip effect.

Although human behaviour in the schematic (vertical) beer distribution game has received considerable attention in academic literature, this is unfortunately not the case for the even more multifaceted topic of horizontal collaboration. Importantly, whereas the beer game covers a fixed four-player interaction (all players are 'in'), under horizontal collaboration, it entails coalition formation from a large pool of potential collaboration partners (some players will be "in", others will be "out"). This results in a completely different structure of interdependence and different behaviour. Understanding how this different behaviour affects the effective design and management of horizontal collaboration initiatives is the main goal of this paper. Therefore, in the next section, we will discuss some key human behavioural factors that are of special importance to horizontal collaboration in transport.

## 3. Results

In this section we provide the main results of our work, i.e., a collection of topics from social psychology that we hypothesise to have an important impact on the success of horizontal collaboration projects. Per topic, we provide a brief description of the social psychological concept to convey the core idea, and then discuss how it relates to horizontal collaboration in transport.

The field of social psychology encompasses a vast body of theory and empirical evidence that can help in exploring the cognitive biases and decision processes of those involved in a multi-agent decision making processes [17]. In this section, we provide a collection of concepts from social psychology that may impact the success of horizontal collaboration initiatives beyond the quantitative savings that are identified by the means of quantitative models. Given the vast field of social psychology, we, by no means, have the ambition to be complete. Instead, we focus on several key social psychological concepts that we consider especially important when trying to understand why collaboration initiatives succeed or fail. Our main goal is to inspire quantitative modelers to consider behavioural aspects thoroughly when designing models or interpreting results. In addition, we focus on the early phases of a developing strategic collaboration, where aspects such as partner selection, rules for profit sharing and mental shift are important. The effective management of running collaborations is out of scope for this paper.

The topics visited are grouped into the following three subsections:

- The individual;
- The group;
- Negotiation.

### 3.1. The Individual

3.1.1. Personality of the Decision Maker

Personality is defined as the "psychological qualities that contribute to an individual's enduring and distinctive patterns of feeling, thinking, and behaving" [18]. The most widely used theoretical framework to characterise personality is the so-called "Big Five" [19]:

- Openness to experience (inventive/curious vs. consistent/cautious);
- Conscientiousness (efficient/organised vs. extravagant/careless);
- Extraversion (outgoing/energetic vs. solitary/reserved);
- Agreeableness (friendly/compassionate vs. critical/rational);
- Neuroticism (sensitive/nervous vs. resilient/confident).

There is a consensus regarding the Big Five model as a unified conceptual framework for personality [20], and it has been the basis for a large body of literature in psychology, where it has been used to explain differences among persons in different areas of inquiry such as leadership, stress and coping, professional success, athletic performance, geopolitical conflict, etc. For example, Palmer et al. [21] find that the personality of the Chief Executive Officer (CEO) is a predictor for firm performance. By extension, this may also hold for individuals who manage a collaboration of multiple companies.

Personality traits are predictors for professional success, ability of conflict resolution, group dynamics, etc. Therefore, especially in strategic collaborations that are designed to last for longer periods of time and depend on a considerable level of trust, it is worthwhile to know the personality traits of the key individuals who represent the various companies in a collaboration. For example, in a study of two of the Big Five personality traits (agreeableness and extraversion), Stavrova et al. [22] demonstrated that, when faced with the dilemma of trust, people prefer agreeable (but not extraverted) others.

In horizontal collaboration, therefore, next to the level of potential savings in terms of cost and carbon emissions, the personalities of the key persons in the collaboration are also of importance for the success of a collaboration. As it is difficult for lay people to assess others' personality traits and the effect that these will have on their trustworthiness, here lies an important role for an independent intermediary or 'trustee' to compose strong

collaborative groups of company representatives who stimulate each other and will be able to solve conflicts without the collaboration falling apart.

### 3.1.2. Social Identity Theory

The social identity theory states that members of separate groups think and act more positively about each other when they think of themselves in terms of a superordinate group identity, i.e., one that they share. Ambrose et al. [23] study this phenomenon in the context of different functional units within a company that must collaborate to enable a smooth Sales and Operations planning (S&OP) process.

For their context, Ambrose et al. [23] define superordinate identity as "the extent to which team members identify with the group, are committed to overarching goals of the group, and have a stake in the collective success or failure of the group". Their results confirm the importance of superordinate team identity in achieving S&OP performance.

The social identity theory posits that people tend to classify themselves and others into various social categories as a means of determining self-identity and belonging to some larger aggregate [24]. Groups with which one perceives association are called "in-groups", while outside and in some cases, competing groups are referred to as "out-groups". As people seek to maximise their self-esteem, they will form positive conceptions of their in-groups and, in turn, they will form negative biases towards out-groups [25].

In organisations, employees tend to form strong identities with their functional unit and negative predispositions towards other units. This tendency is exacerbated by competition for scarce resources and reward systems that typically focus on individual functioning and performance [24]. With horizontal collaboration, in-group favouring to the detriment of the out-group may occur, for example, when only a shortage of transport capacity is available. Having a hidden separation, sometimes referred to as a "fault line" [26] between subgroups in a collaboration project poses a risk to the stability of the entire consortium. When individual or functional goals do not compete with the overall team goals or the performance of the team relies mostly on the aggregate of individual work, superordinate commitment may be less important. However, when there are strong fault lines such as conflicting goals between subgroups, superordinate commitment will be more salient [27].

It is therefore important to look for a shared, or superordinate, identity across the individual members of a collaborating consortium. Seen from the perspective of a superordinate identity, every member of the group is comparable, which facilitates mutual understanding. This is, for example, prominent in mergers, where sometimes there remains competition among the formerly independent companies for years, instead of the collaboration and synergy that was the intention of the merger. Given the typically strong fragmentation of the transport market, concentration through mergers and acquisitions can be expected, and this makes it important to be aware of emerging fault lines when choosing coalitions and search for superordinate identity based on common goals. For a more elaborate discussion of the social identity theory in the context of mergers, we refer to [28].

### 3.1.3. Intergroup Contact Theory

Intergroup contact has been widely studied in the context of intercultural relations, political conflict and discrimination. Popular opinion about intergroup contact is diverse. Some think that contact between groups only causes conflict; in other words, good fences make good neighbours. Others believe intergroup interaction is an essential part of any remedy for reducing prejudice and conflict between groups [29].

The intergroup contact theory was developed by Allport [7], who already noted the contrasting effects of intergroup contact usually reducing, but sometimes exacerbating, prejudice. To explain these findings, Allport adopted a "positive factors" approach. Prejudice will reduce when four positive features of the contact situation are present: (1) equal status of the groups in the situation; (2) common goals; (3) intergroup collaboration; and (4) the support of authorities, law or custom.

Contact effects from one contact situation are typically generalised to new contact situations. Several studies have shown that reduced prejudice against one outgroup can be even generalised to other outgroups that were not involved in the original contact [29]. Furthermore, not all intergroup contacts show positive outcomes in the sense that they reduce prejudice. Some situations increase prejudice.

In practice, collaboration usually takes place among "usual suspects" in the same group, for example, companies that are in the same industry or are located in each other's geographical vicinity. However, the insights of the intergroup contact theory can be used to encourage alternative groups of companies to meet, share experiences and collaborate despite initial (negative) beliefs that they might have about the other. This may, for example, be the case among urban versus rural companies, small companies versus multinational companies, traditional companies versus start-ups, and among directly competing firms. Prejudice that collaboration is not possible with 'the other' can be reduced by having real contact. Positive collaboration experiences that may come out of this first contact are likely to increase openness for new collaborations. To establish contact within the members of the transport industry, thereby fostering collaboration, industry organisations can play an important role as a safe neutral platform.

### 3.1.4. Theory of Mind

The theory of mind is the ability to attribute mental states to others, such as beliefs, intents, desires, emotions and knowledge. Upholding the theory of mind allows us to understand that others have unique beliefs and desires that are different from our own, enabling us to engage in daily social interaction as we interpret the mental states and infer the behaviour of those around us [30]. A main virtue of the theory of mind is the ability to distinguish between one's own mental representations and those of others, i.e., the understanding that different people can interpret reality differently.

The role that the theory of mind plays in negotiations can be analysed in the context of an ultimatum game. This is a simple two-person game in which player 1 (the proposer) receives a monetary endowment and makes an offer regarding how to divide the endowment between themselves and a second player (the responder). The responder then decides whether to accept or reject the proposer's offer. If the responder accepts the offer, each player receives payment according to the proposer's offer. However, if the responder rejects the offer, both players receive nothing. Economic models that use the premise of rational decision making infer that the responder should accept any offer greater than zero and that the proposer should propose the minimum possible offer to the responder. However, empirical data significantly differ from this prediction. In fact, the modal offer made by proposers in industrial settings is 50/50 and the average offer is approximately 60/40 (proposer 60%, responder 40% [31]).

A large part of the literature on horizontal collaboration uses the cooperative game theory to propose or predict the division of the monetary benefits that arise from the collaboration. Rationality is a key assumption in game theoretical allocation rules like the Shapley value, Nucleolus, Tau-value, etc. However, as presented for the ultimatum game, this assumption is violated in experiments. Apparently, the ability to infer the mental states of others, the theory of mind, also plays a key role in fairness-related behaviour. For the exact same collaboration settings, a game theorist will use rationality and selfishness as a premise, whereas a psychologist would start from empathy and perspective-taking (cf. [32,33]). This insight calls for a revision of the gain sharing rules that are commonly used for the analysis of horizontal collaboration.

### 3.1.5. Mindset and Framing

Psychologists have long recognised that understanding, predicting and altering human behaviour requires us to consider the way individuals label the situations with which they are confronted. Liberman et al. [34] conducted an experiment around labelling ma-

nipulation. Their context is the well-known game theoretical situation of the Prisoner's Dilemma, in which participants must decide whether to cooperate or defect.

In their experiment, Liberman et al. [34] varied only the name of the Prisoner's Dilemma game that participants played. For half of the participants, the game was called the "Wall Street Game". For the other half of the sample, the game was called the "Community Game". The former label was meant to connote rugged individualism, self-interest and exploitative norms. The latter label, by contrast, connotes interdependence, collective interest and collaboration. The results showed that in the "Community Game", about 70% of participants showed collaborative behaviour, while in the "Wall Street Game" this was only about 30%. In game theory, this effect is known as the presentation effect [35].

These results suggest that the way (potential) collaborators are primed can make them more open or hesitant to collaborate. Therefore, in the light of the experiment above, it is important to label collaborative efforts in a positive way. A "joint initiative on clean transport" might have a better start than a project labelled "horizontal collaboration in transport".

*3.2. The Group*

3.2.1. Social Dilemmas

The APA Dictionary of Psychology defines a social dilemma as a situation that creates a conflict between the individual and collective interest, such that the individual obtains better outcomes following strategies that, over time, will lead to suboptimal outcomes for the collective. Such situations have reward structures that, in the short run, favour individuals who act selfishly rather than in ways that benefit the larger social collective. Social dilemmas are inherently challenging situations, because acting in one's immediate self-interest is tempting for everyone involved, even though everybody would benefit from acting in the longer-term collective interest. For a review of social dilemmas, we refer to Van Lange et al. [36].

Social dilemmas can be split into two groups. First, there exist "give-some" dilemmas, where individuals choose to renounce short-term own costs with the effect that the collective will not receive a long-term benefit. Paying taxes can be interpreted as a give-some dilemma. The second type, "take-some" dilemmas, is the opposite situation, where individuals can enjoy a short-term, private benefit at the expense of confronting the collective with a long-term disadvantage. An example of a take-some dilemma is the tragedy of the commons, introduced by [37].

Many of the SDGs constitute social dilemmas, since short-term self-interest is often at odds with the longer-term collective interest of achieving the SDGs. This is particularly apparent for SDG13 on Climate Action. In the context of carbon emissions of freight transport, a take-some dilemma occurs when we consider the total carbon emission budget that the industry is allowed to use under the Paris Agreement as the common resource. Operating a profitable, though not fully loaded, truck by an individual company may bring short-term benefits to this company, but is detrimental to the industry in the longer term, since there will be less carbon emission budget left.

A give-some dilemma in collaborative transport can be observed if we extend horizontal collaboration to the industry-wide transition towards more efficient and clean transport. For example, one of the key facilitators for an industry-wide adoption of battery electric trucks is an open charging network. If a company invests in a charging station and then makes it available to others, it incurs the investment costs, while others who do not still benefit. The fear of free riding is an important aspect of give-some dilemmas.

It is clear that social dilemmas can significantly slow down import transitions. Social psychological research shows that the best way to overcome this is a combination of carrots and sticks. It has also been found that such policies of reward and punishment may be even more effective when administered by fellow members facing the social dilemma, rather than by authorities [38]. Ostrom [39] finds that the worst-case scenario when dealing with

social dilemmas occurs when outside parties (e.g., authorities) impose rules on individuals or companies, but then only achieve weak control and enforcement of those rules.

In the absence of strict enforcement, social norms may provide an alternative way to overcome the rational choice of self-interest under social dilemmas. For a description of how social norms can guide behaviour in specific social dilemma situations, we refer to the work of Biel and Thøgersen [40].

### 3.2.2. Social Value Orientation

One of the ways to understand how people behave in social dilemma situations is by measuring people's preferences on the underlying motives that constitute social dilemma situations. The Social Value Orientation construct represents concerns that people have about others' welfare relative to their own welfare. Within the Social Value Orientation construct, people can be categorised into "pro-socials" (individuals who aim at maximising overall benefits and wish to achieve an equitable distribution thereof) and "pro-selfs" (individuals who try to maximise their own welfare either absolutely or relatively). The Social Value Orientation explains how pro-socials tend to collaborate more than pro-selfs [41].

In the context of horizontal collaboration, this makes a case for stressing the ultimate societal goal of collaboration: reducing the overall carbon footprint of transport. Since pro-socials will be more inclined to act according to the group's interest, governments can focus on stimulating pro-socials to be the first movers towards horizontal collaboration to achieve clean and efficient transport. This might be a more effective approach than, for example, an across-the-board subsidy on zero-emission trucks.

### 3.2.3. Discontinuity Effect

The question of how inter-individual behaviour may differ from intergroup behaviour has been investigated broadly in social psychology literature (see, e.g., [42]). A main concept in this literature is the discontinuity effect, which states that conflicts and competitiveness between groups are more intense than between individuals. There are three main reasons to explain this inter-individual intergroup discontinuity effect. First, the "social support explanation" proposes that group members can provide mutual support for the pursuit of self-interest, whereas such social support is not available to individuals. Second, the "identifiability explanation" proposes that intergroup interactions are more competitive than inter-individual interactions, because groups provide a shield of anonymity facilitating the pursuit of self-interest. Finally, the "schema-based distrust explanation" proposes that the anticipation of interacting with another group activates an outgroup schema, consisting of learned beliefs and expectations that intergroup interactions will be aggressive and competitive.

The discontinuity effect might be one of the reasons why horizontal collaboration initiatives often do not come to fruition. Most collaboration initiatives are still a result of informal contact between two or more individuals from separate companies who realise that they can help each other become more efficient together. However, when this collaboration is initiated at their respective companies, co-workers will be included in the project and, from that moment on, the discontinuity effect might raise impediments for the collaboration, which would be easily overcome among the individuals who initiated the collaboration.

### 3.3. Negotiation

### 3.3.1. Sharing

When it comes to sharing benefits and risks in a collaboration, Tyler et al. [43] show that people care about both the actual outcome of a negotiation, but also about the process that led to this result. In other words, fairness perceptions depend on both the self-interest bias ("it is good if it serves me" [44]) and on the perceived process justice [45].

In the field of horizontal collaboration in transport, the sharing of costs and benefits is perhaps the most studied topic. Guajardo and Rönnqvist [46] prepared a separate literature review of this topic, covering 55 papers. Indeed, mistrust about the fairness of

the applied allocation rule for savings has caused many horizontal logistics collaboration initiatives between shippers and/or transport companies to marginalise, disintegrate or even fail to start [3]. Guajardo and Rönnqvist [46] identify more than 40 different cost-allocation methods, which can be categorised into game-theoretical rules, ad-hoc rules and proportional rules. A simple approach for cost allocation is to use proportional allocation that can be based on the overall volume or weight of the products transported. An experiment by van Beest et al. [47] on coalition formation revealed that the use of proportional allocation rules is indeed widespread and that a chosen allocation rule may have an impact on which coalition is formed. Compared to proportional rules, the more advanced approach is to use the cooperative game theory, e.g., by Krajewska et al. [48], but this poses the risk that collaborators support the outcome less, because they might not fully understand the workings of the game theoretical gain sharing rule, resulting in low perceived process justice.

### 3.3.2. The Information Dilemma

In real-life negotiations, negotiators usually have private information, which may affect both the bargaining process and its outcomes. Van Beest et al. [49] studied the psychological effects of having and revealing private information about the payoffs in multi-party negotiations. Negotiators must choose whether they will share their private information truthfully or not. Although revealing private information may help create positive outcomes, because it may help negotiators discover creative outcomes that maximise the joint benefit, a negotiator may risk non-reciprocity and exploitation in more competitive negotiations by revealing information. These dynamics are embodied by the "information dilemma" [50]: revealing information facilitates the achievement of better joint outcomes, but simultaneously increases one's vulnerability.

Experiments by van Beest et al. [49] with three-player negotiations showed that negotiators who were motivated by self-interest rather than by fairness tended to actively conceal their advantage, they made more selfish offers, and they did less well for themselves as a result. In contrast, negotiators who had stronger honesty concerns tended to reveal their advantage, made better offers and eventually did better for themselves. The information dilemma shows that the widespread belief that people should guard their private information and use it to maximise their own outcomes backfires in practice. In fact, the general stereotype that depicts negotiations as competitive, strategic and stealthy was proven to be wrong.

When managing horizontal collaborations, it is important to be aware of the information dilemma. Data sharing between parties in the supply chain is of fundamental interest, since correct and complete information is essential for effective and efficient transport, and this is obviously even more true in the case of horizontal collaboration. However, it is likely that at some point, individuals or companies within a coalition have private information that may affect collaboration benefits for the group, especially in the process of negotiation about the allocation of benefits realised by the group. The information dilemma advocates for open-book accounting in horizontal collaboration projects, showing, for example, not only the predicted, but also the realised outcomes of a collaboration.

### 3.3.3. Strength Is Weakness

Another observation in coalition formation is that the "strongest" negotiators, for example, those with the most resources, the largest market share or the best-known brand name, are often excluded from coalitions: the strength-is-weakness effect. The first mention of the strength-is-weakness effect is found in the work of Caplow [51] on coalitions in a triad. Caplow proposed that strong members try to dominate weaker members. But when the combined strength of the two weaker members was sufficient to control the strongest member, the two weak members would form a coalition against the strong member.

There are two possible explanations for the strength-is-weakness effect. First, the strong member might incorrectly equate their position of having more resources with a

position of having more bargaining power. A second possible explanation comes from the perspective of the weak members, who believe their weakness is an unfair disadvantage compared to the strong member, who is consequently disqualified as a partner. The result is that weak negotiators tend toward each other.

For horizontal collaboration, the strength-is-weakness effect means that size does matter. In mixed groups of companies that consider entering a collaborative arrangement, it predicts that the smaller companies feel more attracted to each other than to the larger company. This does assume, however, that the value of the collaboration is not affected by whether partners are big or small. This assumption may hold for certain situations, but there are also cases where the inclusion of the big company in the consortium disproportionately increases the value of the collaboration. One example could be that small companies can make use of the big player's better purchasing conditions. However, even in such a case, it is important to keep in mind that small companies still feel uncomfortable if the bigger player tries to monetarise on this by negotiating harshly for a large share of the collaborative benefits, which in the long term poses a risk to the collaboration.

### 3.3.4. Trust and Perspective Taking

Trust is a universally valued quality of interpersonal relations, irrespectively of whether these relations are private, professional, economic or political. It can be defined as the willingness of a party to be vulnerable to the actions of another party based on the expectations that the other will perform a particular action important to the trustor, irrespectively of the ability to monitor or to control that other party.

Little is known about how exactly trust can be fostered, but a potential antecedent for trust is perspective taking, i.e., the ability to intuit another person's thoughts, feelings and inner mental state. Erle et al. [33] show through experiments that spontaneous self-reported trust in first-time interactions with a stranger is increased after (visuospatial) perspective-taking. Galinsky et al. [52] found that perspective-takers were better able to uncover underlying interests and can generate creative solutions and better negotiation deals with greater collective and individual gain than control groups in their experiment.

Trust among partners in a horizontal collaboration plays an important role in achieving a successful outcome [53]. Relying on a partner who, in principle, has deviating objectives is a risky undertaking, and therefore trust is necessary to reach a stable form of collaboration. Rindfleisch [54] states that horizontal collaboration among competitors (sometimes referred to as co-opetition) increases the threat of opportunism and lowers the level of trust, because one participant may use information gathered in the collaboration to improve their market position at the expense of others.

Therefore, trust alone is usually not a suitable governance mechanism for horizontal collaboration. Instead, it is advisable to establish a set of collaboration rules in a contract, partially replacing trust with control as a governance mechanism. In addition, company visits, joint events, etc., as a proxy of perspective-taking, may also increase trust.

### 4. Discussion and Conclusions

To comply with the Paris Agreement, alongside technological innovation and a possible transport demand reduction, it is unavoidable that transport companies strongly improve their efficiency and load factors. Furthermore, this seems impossible with the current levels of fragmentation in the industry. Therefore, we expect a transition towards a more collaborative and integrated transport sector.

OR models have often been used to estimate the expected savings of horizontal collaboration initiatives. These models typically find double-digit savings percentages in terms of cost and emissions, and implementation of the collaboration therefore seems an obvious choice. Nevertheless, somehow those who are supposed to make the implementation work in practice do not always act accordingly. In this paper, we conjecture that this can be understood by an omission of important behavioural aspects in most quantitative models for horizontal collaboration. In fact, social psychological insights on mixed-motive interactions

are pivotal for a comprehensive analysis of horizontal collaboration and should therefore receive more attention in the OR literature.

With this paper, we aim to raise awareness of the role of human behaviour and hope to instigate a new stream of research that includes, already from the beginning, social psychological concepts in the design of the collaboration. This can reduce the effort spent on achieving collaborations that seem promising in theory, but that for behavioural reasons are unlikely to succeed in practice.

More generally, to contribute to improved transport efficiency, and hence to the reduction in carbon emissions and climate change, we encourage further interdisciplinary research into horizontal collaboration. In this paper, we have shown that important concepts from social psychology that play a role in collaboration and coalition formation are commonly not modelled in OR. We hope to inspire academics from both fields to learn from each other, since achieving SDG13, as is the case for all SDGs, will never happen within the confines of a single academic discipline. Our call for interdisciplinary research efforts among OR and social psychology is therefore just one example of the collaborative actions that must be undertaken to help the logistics industry to become sustainable.

There are various directions for future research that will further increase our understanding of (the success of) horizontal collaboration as a way to reduce the environmental footprint of transport. First, we encourage explanatory studies showing how social psychology concepts can be used concretely in certain transport collaboration environments. The first attempt to do this can be found in Cantiani et al. [55], who find that the strength-is-weakness effect indeed exists in a stylised transport collaboration experiment conducted via 1) a generic panel of 567 participants; and 2) various smaller groups of supply chain decision makers. Secondly, empirical research via surveys can shed more light on how logistics decision makers regard psychological aspects.

**Author Contributions:** Conceptualisation, F.C., I.v.B. and G.K.; Methodology, F.C. and I.v.B.; Investigation, F.C. and I.v.B.; Writing—original draft, F.C.; Writing—review and editing, F.C., I.v.B. and G.K.; Project administration, G.K.; Funding acquisition, F.C., I.v.B. and G.K. All authors have read and agreed to the published version of the manuscript.

**Funding:** This study was funded by a grant from the Dutch TKI Dinalog under their program for a multidisciplinary approach of chain collaboration, with file number 439.16.604.

**Institutional Review Board Statement:** Not applicable.

**Informed Consent Statement:** Not applicable.

**Data Availability Statement:** Data sharing is not applicable. This paper aims to open a research agenda in the interdisciplinary field of social psychology and Operations Research, with the goal to foster impact-based research that contributes to the achievement of sustainable transport. Therefore, no new data were created or analysed in this study. Rather, the paper provides a novel combination existing literature of the two separate fields.

**Acknowledgments:** The authors wish to heartfully thank Bas van Bree of TKI Dinalog for encouraging us to investigate the role of human behaviour in the transition towards sustainable transport.

**Conflicts of Interest:** The authors declare no conflict of interest.

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
