# Peer review of "A Human Behaviour Perspective on Horizontal Collaboration to Reduce the Climate Impact of Logistics"

_sustainability, doi:10.3390/su152316221_

Round 1

Reviewer 1 Report

Comments and Suggestions for Authors

I appreciate your submission to the Journal of Sustainability. While your paper contains intriguing concepts, it requires significant revisions before it can be considered for publication. Additionally, I recommend thorough proofreading.

In reference to page 2, line 67, where the authors mention "emphasis on methodological analysis," further elaboration is needed to clarify the significance of this approach.

Regarding page 3, line 99, the overarching aim of the paper should be detailed more comprehensively. Specifically, the authors should elucidate their rationale for introducing a model for human behaviour and its potential to enhance sustainable capabilities.

Overall: 

Clarifying the Target Audience: The paper lacks specificity and clarity regarding its target audience. It is essential for the authors to refine their focus and tailor the content to better cater to their primary readership. 

Improving Diagram Clarity: The purpose of the diagrams is not clearly conveyed, and the authors should provide more comprehensive explanations for each graph.

Enhancing the Conclusion: In the conclusion section, it is essential to outline the implications of the research for future studies and to transparently discuss the limitations of the current study. Additionally, the significance of this research within the field of sustainability should be prominently emphasised.

Author Response

Dear reviewer,

Thank you very much for taking the time to review our paper, and for your useful comments, which have helped us to improve the quality of the paper. Please find our responses to your comments in the attached file.

Best regards, the authors

Reviewer 2 Report

Comments and Suggestions for Authors

This is an interesting study. Research gap has been identified. But authors need to create a debate so that readers could read the article to the end. Now is only present the issue, HC. Social identity theory, theory of mind and intergroup contact theory were discussed. Looking your conclusion is not surprise to me. Authors need an overarching theory to discuss issue. What are the theoretical contributions?

Comments on the Quality of English Language

English editing is required

Author Response

(The authors gave the same response as above.)

Reviewer 3 Report

Comments and Suggestions for Authors

Dear Authors,

The manuscript presents a quite interesting study, but for me some aspects must be improved:

-        In the introduction part maybe you could ”introduce” the reader in the researched subject, with a short paragraph, not start directly with subchapter. In my opinion I would renounce at all at all those 3 subchapters from this part.

-        The manuscript does not accomplish the structure mentioned by the journal: introduction, material and methods, results, discussion and conclusion…. but the worst thing is that it does not have the results and discussions part!!!!

-        The manuscript is hard to read and understand in this format…so it must be reorganized!

-        The implications and limitations of the study are too short (almost absent). Hence, the authors should add implications (both practical and theoratical) and future directions of the research.

Author Response

(The authors gave the same response as above.)

Reviewer 4 Report

Comments and Suggestions for Authors

The manuscript - A Human Behaviour Perspective on Horizontal Collaboration to Reduce Climate Impact of Logistics - is a beautifully written article. I enjoyed reading this paper. This paper is about reducing carbon footprints from the transport sector with horizontal collaborations as defined in line 49 - a collaboration between two or more firms that are active at the same level of the supply chain and perform a comparable logistics function. The article is based on a literature search and aims to establish an interdisciplinary research agenda that encourages the quantitative research community studying horizontal collaboration to explicitly model for human behaviour to enhance the logistics and supply change organizations' sustainable capabilities. However, the agenda implicit in this article is not explicit to the general audience. I, therefore, recommend some possibilities to improve this manuscript. 

1. The agenda proposed in this paper would better come not only in words but also in a figure, a mind map, or a framework structuring all the elements of the agenda proposed. Only this way, it merits as an article otherwise looks just like a review article.  

2. Lines 55-58, where did the authors search keywords? the number of total references found? The quality of the Figures is poor. 

3.  Minor formatting issues include citation styles are not according to the journal guidelines. 

I look forward to reading the improved version of this article. 

Author Response

(The authors gave the same response as above.)

Reviewer 5 Report

Comments and Suggestions for Authors

This work is interesting. In addition to human behaviors, we must consider how political actions impact climate change. For instance, wars can result in significant expenditure of resources such as oil, gas, and energy, transforming forests and residences into ash. This constitutes a crucial factor. A single regional conflict can play a devastating role in escalating the risk of climate change. How can we, as humans, work to prevent this? The authors have also identified minor revisions that will enhance the submission.

Figure 1 is unclear, and the labels for the x and y axes are missing.

In Section 1.3, please revise the paragraph in an objective manner, avoiding the use of first-person pronouns.

Figure 2 is also unclear and requires clarification.

On Line 255, it is suggested to consider political and war-related factors. These elements serve as pivotal drivers for exceeding carbon dioxide levels.

In Section 3, in addition to your discussion of individual, group, and societal awareness, could you also provide a formulation of these factors along with corresponding numerical results? This enhancement is crucial for advancing social studies.

Comments on the Quality of English Language

Moderate editing of English language required.
Please revise the paragraph in an objective manner, avoiding the use of first-person pronouns.

Author Response

(The authors gave the same response as above.)

Round 2

Reviewer 1 Report

Comments and Suggestions for Authors

Thank you for taking the feedback into account. The paper is now ready for publication.

Author Response

Thank you very much for your time invested in reviewing our paper, and for your positive assessment.

Best regards, the authors

Reviewer 2 Report

Comments and Suggestions for Authors

This is a good start for the subject. Authors need an overarching theory to discuss the issue. Now is too board from the perspective of social psychology. The next step of the research is to design an explanatory study. Study 2 could be a large scale survey. Those studies might validate your argument. Good Luck!

Comments on the Quality of English Language

Some improvements.

Author Response

Dear reviewer,  we are a happy to read that you acknowledge the importance of the subject and propose two additional studies as a follow-up to the foundational work that we do in the current paper.

In fact, this is exactly what we want to achieve with our paper: starting an 

In fact, this is exactly what we want to achieve with our paper: starting an interdisciplinary research field with input from supply chain management, optimization and social psychology, so that we will be able to better understand why logistics certain optimization and collaboration projects succeed, while others fail. As argued in the paper, this is of vital importance to achieve the stringent climate goals for the transport industry. We support your two suggestions, i.e. an 1) example study of how social psychology concepts can be used in a more narrow and concrete application (‘explanatory study’, in the words of reviewer 2), and 2) a survey research to map to shed light on how logistics decision makers regard psychological aspects (‘large scale survey’). These are two good suggestions that are natural follow-ups of our paper, so we have described them as directions for future research in our final chapter, as follows:

Page 13: There are various directions for future research that will further increase our under-standing of (the success of) horizontal collaboration as a way to reduce the environmental footprint of transport. First, we encourage explanatory studies showing how social psy-chology concepts can be used concretely in certain transport collaboration environments. A first attempt to do this can be found in Cantiani et al. (2023), who test both the effect of perspective taking and of the strength is weakness effect in a stylized transport collabora-tion experiment conducted via 1) a generic panel of 567 participants and 2) various smaller groups of supply chain decision makers. Secondly, empirical research via surveys can shed more light on how logistics decision makers regard psychological aspects, see for example Kant et al. (2021) for a first exploration.

Reviewer 3 Report

Comments and Suggestions for Authors

Dear authors

I apreciate your effort to improve your research but you did not follow all the requirements!!!!

-        The manuscript does not accomplish the structure mentioned by the journal: introduction, material and methods, results, discussion and conclusion…. but the worst thing is that it does not have the results and discussions part!!!!

Author Response

Dear Reviewer 3, thank you  for your second review of our paper. We asume that you are addressing a formatting point, namely to use the standard Sustainability journal section structure of 1) introduction, 2) materials and methods, 3) results and 4) discussion. This remark was also made in the first review round, and we formulated the following response:

“Before submitting our original version of the paper, we carefully considered the best structure (section names) of it and eventually decided to deviate from the standard journal structure to which you allude. The reason is that the purpose our paper, i.e., instigating a new interdisciplinary research stream, does not match the outline of a traditional research paper that would have an introduction, method, results and discussion structure. Nonetheless, we appreciate your concern and believe it resonates with the overall issue that we should explain better what our contribution is and what our target audience is. We hope we clarified this by rewriting the introduction and also general discussion.”

We understand that this answer does not fully satisfy you and therefore we have now restructured our paper using the four standard sections as requested. To accommodate this, we have added two additional fragments:

Page 3: In this sections we provide the preliminaries to explain how our study builds upon earlier published literature. All literature that has been used can be found in the list of citations at the end of the paper and there are no specific restrictions to report on the availability of this material.

Page 6: In this section we provide the main results of our work, i.e. a collection of topics from social psychology that we hypothesize to have an important impact on the success of horizontal collaboration projects. Per topic we provide a brief description of the social psychological concept to convey the core idea, and then discuss how it relates to horizontal collaboration in transport.

Reviewer 4 Report

Comments and Suggestions for Authors

The authors have significantly improved the quality of work according to the given suggestions and it looks fine from my side. 

Author Response

(The authors gave the same response as above.)

Reviewer 5 Report

Comments and Suggestions for Authors

Thanks to the author for the revision. The paper is now in a much more comprehensive version and is recommended for publication.

Author Response

(The authors gave the same response as above.)

Round 3

Reviewer 2 Report

Comments and Suggestions for Authors

Some improvements made. References are updated. A paragraph on future research was added on Discussion section. The sub-titles was used to make the flow better in the paper. Sorry, authors do not address my previous comment in full.

Comments on the Quality of English Language

Improvements made.

Author Response

Dear reviewer,

Thank you for your comments. You indicated: “Some improvements made. References are updated. A paragraph on future research was added on Discussion section. The sub-titles was used to make the flow better in the paper. Sorry, authors do not address my previous comment in full.”

We agree with your evaluation that we took the paper as far as we could. We have been able to update the references and have been able to clarify the goal of our paper. In doing so, we also followed your advice to be clear on what type of follow up-research could be done. Moreover, we also included some examples of research that already benefitted from integrating social psychology and operations research. We understand from your final comment that you had hoped that we would have been able to push this even further. We respectfully disagree. We think we have reached a saturation point. The goal of our paper is to provide a research agenda for the future. The goal of our paper is not to already validate this research agenda with empirical research in this paper.

Reviewer 3 Report

Comments and Suggestions for Authors

Dear authors the discussion part must be the discussion and conclusion. 

Author Response

Dear reviewer, thank you for the following comment:

“Dear authors the discussion part must be the discussion and conclusion.”

We have changed the title of the final section to “Discussion and Conclusions”.

Thank you very much for taking the time to review our paper in various rounds, we have happy that we have been able to accommodate your feedback in the revised version of our paper.